# A New Miniaturized Gas Sensor Based on Zener Diode Network Covered by Metal Oxide

**DOI:** 10.3390/mi12111355

**Published:** 2021-11-02

**Authors:** Vignesh Gunasekaran, Soffian Yjjou, Eve Hennequin, Thierry Camps, Nicolas Mauran, Lionel Presmanes, Philippe Menini

**Affiliations:** 1CNRS, LAAS-MICA, 7 Avenue du Colonel Roche, 31400 Toulouse, France; vgunasek@laas.fr (V.G.); syjjou@laas.fr (S.Y.); ehennequ@laas.fr (E.H.); camps@laas.fr (T.C.); nicolas.mauran@laas.fr (N.M.); 2CIRIMAT, CNRS-INP-UPS, Université Toulouse 3 Paul Sabatier, 118 Route de Narbonne, CEDEX 9, 31062 Toulouse, France; presmane@chimie.ups-tlse.fr; 3Faculté des Sciences et Ingénierie, Université Toulouse III Paul Sabatier, 118 Rte de Narbonne, 31400 Toulouse, France

**Keywords:** miniaturized gas sensor network, p-n junction, metal oxide, RF-sputtering, ZnO:Ga thin layer

## Abstract

The development of “portable, low cost and low consumption” gas microsensors is one of the strong needs for embedded portable devices in many fields such as public domain. In this paper, a new approach is presented on making, on the same chip, a network of head-to-tail facing PN junctions in order to miniaturize the sensor network and considerably reduce the required power for heating each cell independently. This paper is about recognizing a device that integrates both sensing and self-heating. This first study aims to evaluate the possibilities of this type of diode network for use as a gas sensor. The first part concerns the description of the technological process that is based on a doped polysilicon wafer in which a thin layer of metal oxide (a gallium-doped zinc oxide in our case) is deposited by RF sputtering. An electrical model will be proposed to explain the operation and advantage of this approach. We will show the two types of tests that have been carried out (static and dynamic) as well as the first encouraging results of these electrical characterizations under variable atmospheres.

## 1. Introduction

The current sensors, both commercial and those developed in lab, are the subject of compromises: size, sensitivity, selectivity, consumption and cost [1]. The optical sensors by direct infrared absorption remain the most efficient and most selective, but they only address one target gas. However, their size and high power consumption are unfavorable to realize embedded multi-gas sensors. Electrochemical sensors, widely present in the market due to their good performance in sensitivity and selectivity, remain penalized by their size, their significant consumption, and especially, by their limited lifetime. For their part, electromechanical sensors (MicroElectroMechanical Systems (MEMS), quartz micro balances (QMB) or electro-acoustic sensors (bulk acoustic wave (BAW), surface acoustic wave (SAW)) reveal excellent sensitivities and low detection thresholds (at ppb level) but require a complex electronic system for driving and measuring—not easy to integrate in simple embedded devices. Finally, the conductometric sensors still remain relevant for integration of multi-gas detectors in a reduced volume, moderate power consumption, and above all, simple electronics as an interface with good reliability [2]. Due to significant advances in sensitive materials (nano-structured metal oxides, inorganic materials, etc.), these sensors now display strong sensitivities and low detection thresholds (several tens of ppb).

More specifically, microsensors based on microhotplates composed of hanging membranes with sensing electrodes and heating elements have proven its high application to various gas-sensitive elements with low power consumption [3,4,5]. This kind of platform (Figure 1) is compatible with various technics of sensing layer integration, such as inkjet printing or microplotter printing [6,7], screen-printing [8,9] or magnetron sputtering [10].

Despite the tiny size and the ease of fabrication, due to its smart structure, it takes more than five photolithography processes to pattern all the different layers, including a backside deep reactive ion etching (DRIE) to obtain the suspended round membrane, adding, to the whole system, a mechanical weakness [11]. The heating element was designed to give a homogeneous temperature distribution across the membrane to bring the sensing layer to the working range. However, the thermal image of the microhotplate showed that only a small region in the center of the membrane is at the required temperature [12], and there will be always a significant temperature gradient that is not acceptable for the whole sensing material at the surface.

Concerning the sensing electrodes, we can find few different shapes that are usually suited to the sensitive material resistivity. Some can be a miniaturized pair of electrodes with few microns as a gap [5] or interdigitated electrodes, as an example: the comb shaped sensing electrodes given in Figure 2 [13] with tight interdigitated electrodes. The gap between the two electrodes is L = 10 µm, and the probed length W = 1620 µm gives a number of square n_□_ = L/W of 6.2 × 10^−3^□, and a sheet resistance (R_□_) in a range from 10 kΩ/□ to 10 GΩ/□. If we consider the thickness range of the sensing layer (around 100 nm), we can calculate the range of resistivity that suits this platform. In this case, we have the possibility to measure materials with a resistivity from 0.1 Ω·cm to100 kΩ·cm. To be able to characterize materials with higher resistivity, which is usual for various metal oxides (MOX), it could be interesting to significantly reduce the gap L between the two electrodes.

The weak point of these “resistive” sensors remains their low intrinsic selectivity related to the resistance measurement influenced by many gases (and other influencing factors), regardless of the type of the material used. This drawback can be overcome by the use of a multi-sensor (with various materials) associated with an optimized operating mode.

However, their use (in case of numerus sensors in package) in an autonomous portable system remains limited by their power consumption due to the heating resistor (several tens of mW), which is necessary for the variability of the sensitivity and the reversibility of the sensor. This indicates the importance to develop new miniaturized selective gas multi-sensors with low power consumption (<1 mW) and a low limit of detection (several ppb).

## 2. Zener Diode-Based Platform

### 2.1. Design and Model of the Device

We propose here a new type of gas-sensing platform based on an ultra-miniaturized matrix gas multi-sensor with multiple Zener diodes in series and parallel powered by a single source. This concept is based on a simple and robust technological approach, such as the locally doping polysilicon layer, and then it uses a powerful transduction technique (with a simple current measure), and consequently an ease-of-use electronic interface. Above all, this approach allows low power consumption (due to the small reverse current in a diode) well suited for applications focused on embedded systems and Internet of Things.

As can be seen hereafter, it consists of a simple technological process flow that can be summarized in only three photolithography steps. From initial p-type polysilicon layer, by locally doping this polysilicon (to become a n-type region), we obtain a double junction that is head-to-tail (Figure 3b). When one is in forward conduction (V_D_), the other will be in reverse bias; thus, in Zener conduction (V_Z_), the overall conduction will be driven by the diode in reverse conduction. Due to an insulated SiO_2_ layer opened above only one junction, we can define a reference diode (covered by SiO_2_) and a sensing diode (covered at the end by a metal oxide layer) (Figure 3c).

Thus, if a current is applied between terminals A and B, the reference diode is in Zener conduction, and the evolution of the overall voltage of the sensor will be almost the same as that of the single reference diode in Zener conduction. However, if the polarity of this current is reversed, the measured voltage V_AB_ will almost coincide with that at the terminals of the sensing diode. Thus, by simple subtraction of the forward and reverse currents, the conduction in the junctions is removed to keep only that linked to the conduction in the MOX layer deposited on the open area above the sensing diode.

Effectively, to obtain a miniaturized gas sensor, a gas-sensitive layer (a metal oxide layer for example) is deposited above the detection diode in order to allow current to pass through this layer instead of through the junction in reverse conduction (see Figure 3c). In addition, due to the high doping level, the space charge zone in reverse polarization is reduced (<1 μm) and corresponds to the inter-electrode distance of conventional interdigitated structures. This opens up prospects for characterizing sensitive layers of high resistivity. The resistances of the lateral access zones (contact resistance and in the N and P polysilicon layer) are negligible due to the high level of doping of the N and P zones (N_D_ > 10^21^ cm^−3^, N_A_ > 10^18^ cm^−3^ respectively).

Furthermore, by increasing the level of current flowing through the sensing diode in Zener conduction, the dissipated electrical power focuses on the sensing area (delimited by W_T_ (Figure 3c)). Thus the localized self-heating can reach several hundred degrees with reduced levels of the applied current (around 1 mA), and play the role of self-heating element in this new type of platform. Thus, the significant reduction in the heating zone should make it possible to reach high temperatures necessary for gas detection in the MOX, with a reduced power of typically below 1 mW and with fast thermal transients (<ms) due to ultra-localized power above the junction (>100 kW/cm^2^). This approach then allows the possibility to modulate the operating temperature of the sensor (well known in classical MOX sensors) by adjusting the diode reverse current, another way to optimize the sensor sensitivity for each sensing material.

### 2.2. Fabrication of the Device

The fabrication of this device uses only basic and standard microelectronic processes. First of all, a thick SiO_2_ passivation layer (2 µm) is deposited by low pressure chemical vapor deposition (LPCVD) onto a standard silicon wafer, followed by deposition of a polysilicon layer also by LPCVD, and then a full wafer boron implantation permit to adjust p-type doping level and local n-type doped zones by phosphorus diffusion through a SiO_2_ mask to obtain a pair of Zener diodes (N++/P+/N++ junctions) (Figure 3a). The whole structure is passivated by a 300 nm SiO_2_ layer (also by LPCVD) (in light green on Figure 3a). Then, the contact zones and the active detection area are opened by RIE etching. Finally, the contacts pads, in aluminum, deposited by sputtering (500 nm), are patterned by wet chemical etching. As the N++ regions of polysilicon are degenerated, the contact is ohmic and has low contact resistance (<1 Ω) (in yellow on Figure 3).

In order to obtain better sensitivity and a large working area, we designed a diode array (matrix) (2 × 2 mm^2^), keeping the head-to-tail facing diode configuration (Figure 4). This first operational platform has a total working length of W = 6.6 cm and a gap of L = 0.5 µm, which gives a ratio of L/W = 7.6 × 10^−6^□ and thus a sheet resistance R□ in the range of 130 MΩ/□ to 13 TΩ/□ corresponding to sensitive MOx resistivity from 10 kΩ·cm to 1 GΩ·cm. This is large compared with standard interdigitated electrodes on microhotplate platforms. Thus, this platform is well adapted for high-resistive materials.

The gas sensing ability of this whole system was tested using, first, a 25 nm thick gallium-doped ZnO (ZnO:Ga). Zinc oxide is already studied for various applications, and its gas-sensing ability makes it interesting for our work. The choice of gallium-doped zinc oxide comes from a previous work, where its gas-sensing performance was proven in a framework of classical resistive sensors [11].

The sensitive layer was deposited by radio-frequency (RF) sputtering. This method is well suited with the industrial fabrication of miniaturized microelectronics. RF sputtering presents many other advantages, such as the possibility of making thin films with nanometric scale grain sizes and controlling, without difficulty, the inter-granular porosity [14,15] by varying deposition conditions [16]. Controlled nanostructures films, such as these, present great interest for their potential gas-sensing ability once integrated onto gas-sensing platforms [14]. Films with a controlled nanostructure are of great interest because of their potential to act as sensitive layers after being integrated onto gas-sensing devices [17].

The thin film was obtained by sputtering, using a homemade sintered ceramic target of ZnO:Ga with a density of around 70%, and the deposition parameters were optimized to promote maximum intergranular porosity. Table 1 presents the whole parameter set of the experiment [18].

A first 25 nm thick ZnO:Ga layer was deposited onto fused silica substrate for morphological characterization using the D3000 VEECO Atomic Force Microscope (AFM). Tapping Mode images were obtained using silicon TESP-SS cantilevers with a resonance frequency of around 300 kHz. The composition of this thin layer was verified by the microprobe technique and the gallium-doping rate was that of 4% by mass.

The AFM observation shows a surface with well-defined circular grain and intergranular porosity (see Figure 5). Using image analysis [19], the mean grain size is centered on 4.8 nm with an error of 0.1 nm, and the measured roughness for this sample is 1.1 nm. This is excellent for interactions with gas molecules. This morphology is typical of a sputtering deposition and is well known from previous studies [20].

This gas-sensitive layer was deposited onto the platform using RF magnetron sputtering through polymer shadow mask, with the correct opening above the diodes. The shadow mask (in kapton) was stuck on the surface of the device, exposing the right area to deposit, and was then removed after deposition. First prototypes were obtained (Figure 6), and the chips were mounted in a TO-8 package with the electrodes, and the package was connected with wedge bonding using 25 µm aluminum wires.

### 2.3. Preliminary Tests and Observations

#### 2.3.1. Role of Doping Level

Various Zener diodes with different p-type initial doping levels was fabricated, and the characteristics of the whole system were first studied (Figure 7).

As shown in Figure 7, the behavior of the reference diode (covered by SiO_2_) and of the sensing diode (with MOX at the surface) for different doping levels of the P + zone is distinct.

The forward characteristic shows, as expected, that the Zener voltage decreases when the doping of the P zone increase. Conversely, the reverse characteristics seem less dispersed and show that the overall conduction integrates a non-negligible contribution through the MOX layer.

To enhance the impact of this conduction in the MOX layer, we opt for the less doped platform (N_A_ = 3 × 10^17^ cm^−3^), which corresponds to the highest Zener voltage close to V_Z_~10 V.

#### 2.3.2. Influence of Surrounding Atmosphere

Figure 8 shows the I-V characteristics of the reference diode and the sensing diode under ambient atmosphere (with 50% of relative humidity) and then under dry air flow in the probe station chamber.

After further tests, it is clear that only the sensing diode threshold voltage (directly in contact with atmosphere) is sensitive to the humidity variation in the air, contrary to the reference diode. That proves the surface effects, by the shift toward the high values of the threshold voltage, of the sensing diode, even at low current.

A similar qualitative test was performed using isopropanol (IPA) vapor injected manually into the chamber to observe the behavior of the diodes and to assure that there is no diffusion of COV through the SiO_2_-insulated layer. For that, two sets of diodes were considered, the first one with a naked sensing diode and the second one with both diodes covered by a SiO_2_ layer. Both sets were exposed to the ambient air and a small amount of IPA vapor.

Figure 9 shows the characteristics of the two sets. The passivated one showed a classical Zener diode behavior, and the naked sensing diode showed different responses to the ambient air and under IPA, proving again, the concept of gas detection due to the naked P-N junction compared with the passivated one.

In this final preliminary test, we wanted to compare the sensing diodes I-V characteristics with and without the MOX sensitive layer. It is obvious (i.e., Figure 10) that the reverse current flow passed through the P–N junction without MOX, revealing high resistance. When the MOX layer is present, the equivalent resistance is significatively lower at low current (100 nA), which proves that the reverse current in the Zener diode is well shunted by the current in the MOX. This is precisely what we wanted.

## 3. Results of Gas-Sensing Tests

### 3.1. Test Protocol

The apparatus was placed in a chamber with a controlled atmosphere and connected to a gas dilution bench, where the gas concentration is controlled by a program under Labwindows CVI, implemented on a computer. It is also possible to control the overall flow from a few cc/min to 1 L/min, and the relative humidity at room temperature can vary from a few % to 70%. The micro-sensor is either connected to a source-measure unit (SMU—NI-PXI) for static measurement or connected to a Hioki impedance analyzer and driven by LabVIEW for dynamic analysis.

For the static test, dry air was continuously injected into the 250 cc quartz chamber, and after two hours, 10 ppm of formaldehyde was injected during 30 min. In a second part, humid air (with 30% relative humidity) was continuously injected, and after 1.5 h, 10 ppm of formaldehyde was again injected into the humid air. The total flow rate was continuously maintained at 200 cc/min. First, the static response of the sensor consisted of measuring the overall resistance when a 1 mA reverse current was injected into the sensing diodes (Figure 11).

In the second step, the dynamic tests consisted of measuring the impedance with a frequency sweep from 10 Hz to 10 kHz and with one sample per second during the same gas injection diagram shown previously.

### 3.2. Results and Analysis

#### 3.2.1. Static Measurements

The first results of the static test reveal a variation of resistance during the two injections of gas as well in the dry air, such as in humid air, proving that this new device reveals a significant response to humidity and gas variations (Figure 12). Despite its low sensitivity (a few % for 10 ppm of formaldehyde), the sensor is also able to detect this target gas in dry air and at 30% of relative humidity, which demonstrates the proof of concept of gas detection at room temperature.

#### 3.2.2. Dynamic Measurements

Figure 13a,b shows the results of the dynamic tests. There is a shift to lower values of impedance when injecting target gas at low frequencies, confirming the results of the static test, and at higher frequencies (at 10 kHz). Although the real part (resistance) reveals small variations, the imaginary part (capacitance) is significantly affected by gas and humidity.

The dynamic test makes it possible to measure two characteristics to distinguish the target gas, which can hopefully be used to improve the selectivity of this new type of gas sensor.

## 4. Conclusions

We have presented a new type of miniaturized gas sensor based on a surface conduction of a Zener diode coated with porous ZnO:Ga thin layer using RF sputtering. This technology allows a high level of integration in terms of the probed surface of the detection material (level less than ppm) and then allows the analysis of conduction in materials with high resistivity. This approach also makes it possible to operate with the reverse bias current: at low level (few µA) during measurement and few mA to exploit self-heating and thus control the sensitivity of the sensor.

We proved the effects of the surface on the characteristics of the diode with and without the MOX layer, which allowed us to use this head-to-tail diode configuration as a platform for gas-detection materials.

Two types of tests were carried out on the use of this micro-platform, a static electrical test and a dynamic electrical test with a frequency sweep from 10 Hz to 10 kHz. The results show a resistance response to 10 ppm of formaldehyde gas in dry air and at 30% humidity. The dynamic test reveals a variation of real part (resistance) at low frequency and a variation of imaginary part (capacitance) at high frequency, which is promising in the perspective of improving selectivity.

Despite the low sensitivity obtained by these first, and not optimal, experimental results, this new device revealed encouraging results, and this device can be used as a gas sensor.

In perspective, to make this new device more reliable and more sensitive, we must carry out more tests under various gases and study the effect of (i) the design (number of diodes in parallel), (ii) the temperature (by playing with different reverse bias current) on the entire structure and of course, and (iii) other sensitive materials to consider the possibility to realize a multigas sensor platform.

## Figures and Tables

**Figure 1 micromachines-12-01355-f001:**
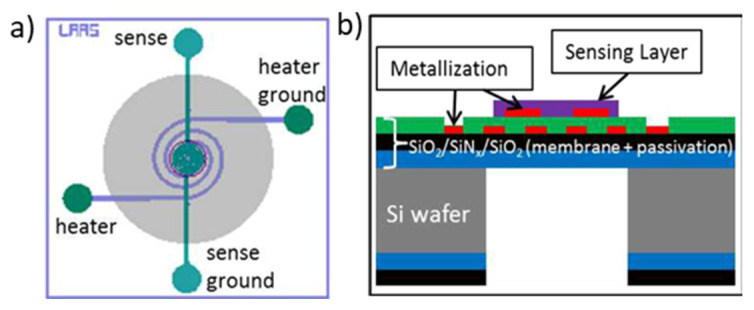
(**a**) Top view of the microhotplate; (**b**) Cross section of the sensor.

**Figure 2 micromachines-12-01355-f002:**
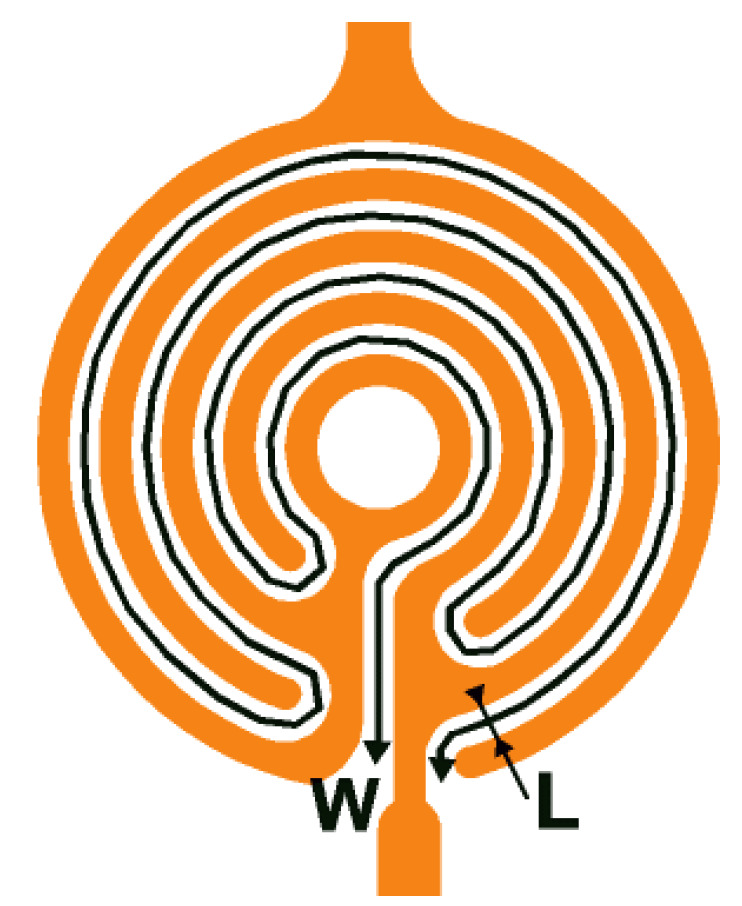
Structure of the comb-shaped sensing electrodes.

**Figure 3 micromachines-12-01355-f003:**
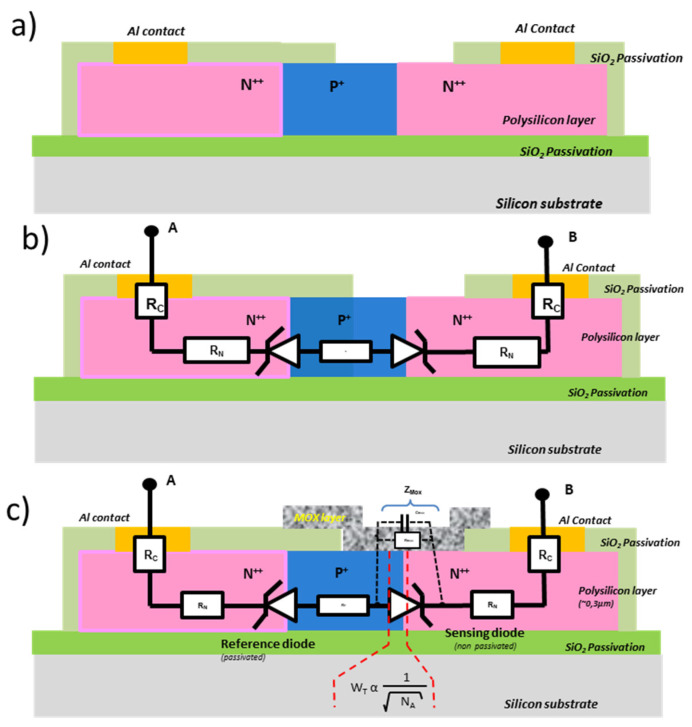
Basic cross section of Zener diode-based platform: (**a**) the overall structure of a single element; (**b**) the electrical equivalent circuit; (**c**) the electrical equivalent circuit after the sensitive MOX layer.

**Figure 4 micromachines-12-01355-f004:**
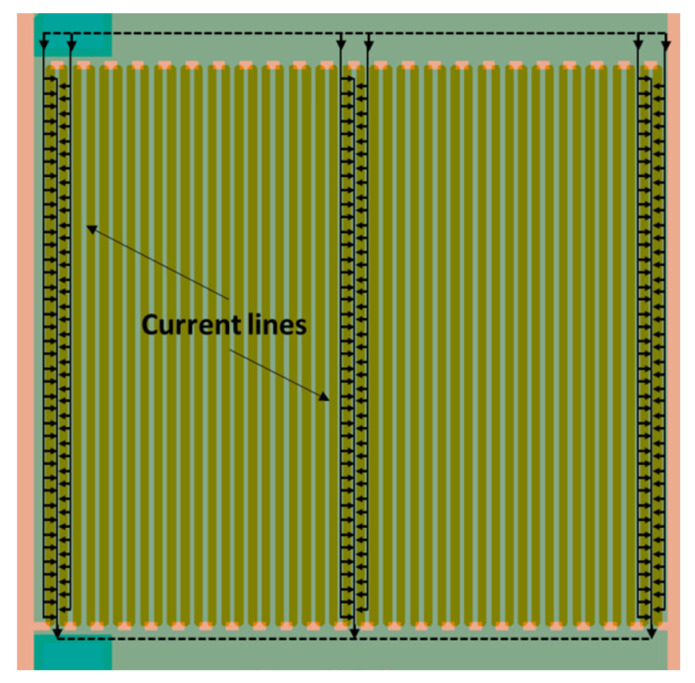
Example of structure of the new sensing platform with 46 pairs of diodes.

**Figure 5 micromachines-12-01355-f005:**
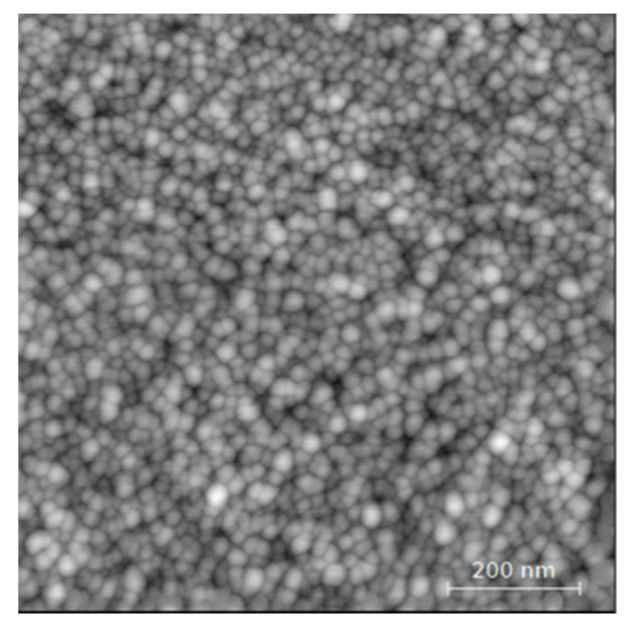
AFM image of a 25 nm thick ZnO:Ga.

**Figure 6 micromachines-12-01355-f006:**
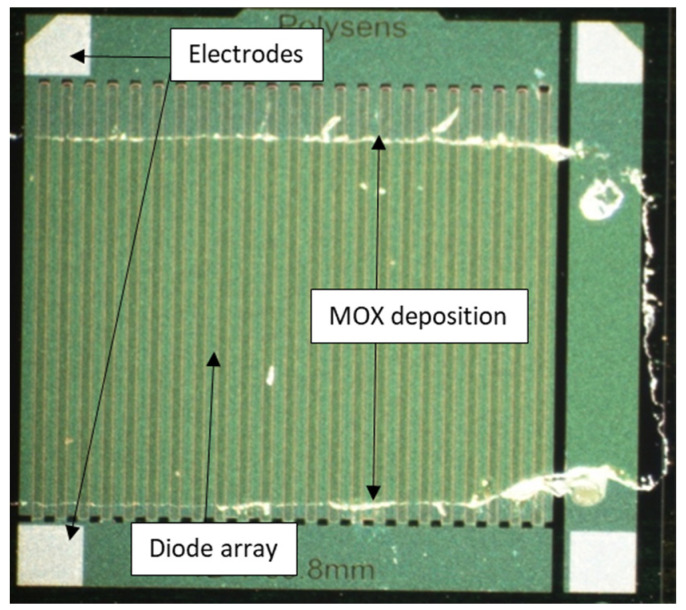
Single device of Zener diode array with MOx thin layer deposition.

**Figure 7 micromachines-12-01355-f007:**
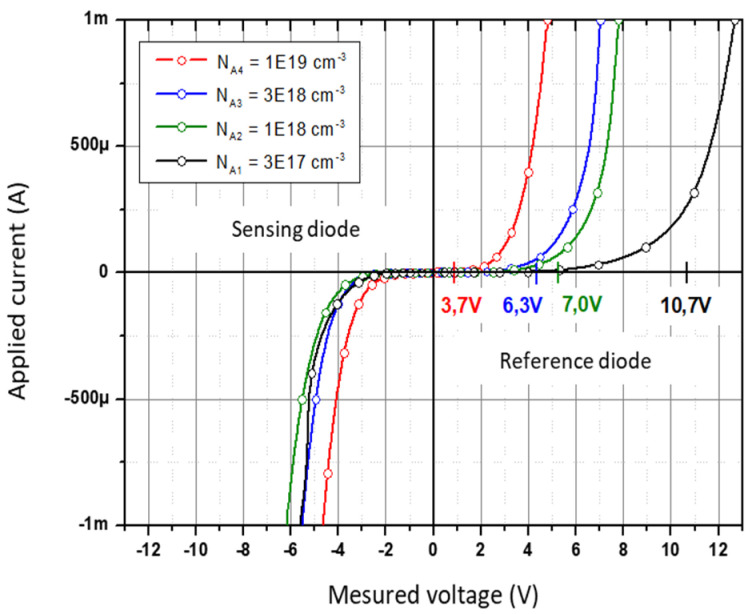
Electrical characterization of pair of Zener diodes with different initial doping level.

**Figure 8 micromachines-12-01355-f008:**
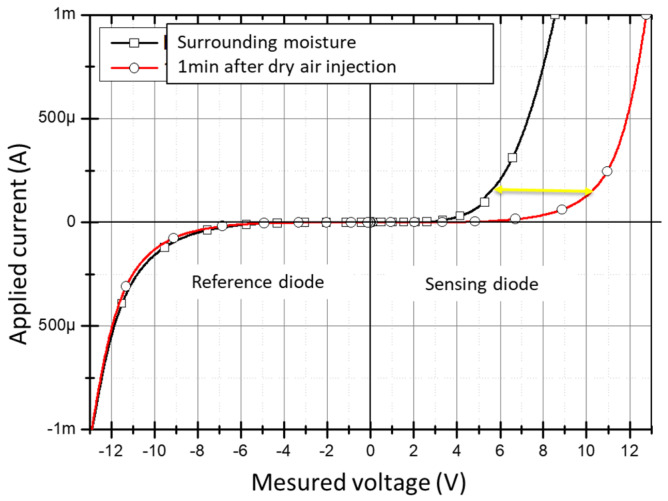
Characterization of the diodes set under ambient atmosphere and dry air flow.

**Figure 9 micromachines-12-01355-f009:**
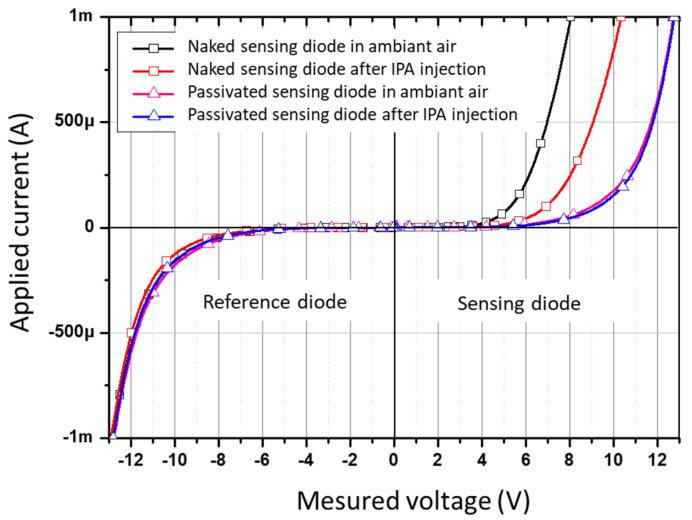
Characterization of two sets of diodes with and without passivation under ambient air and isopropanol vapors.

**Figure 10 micromachines-12-01355-f010:**
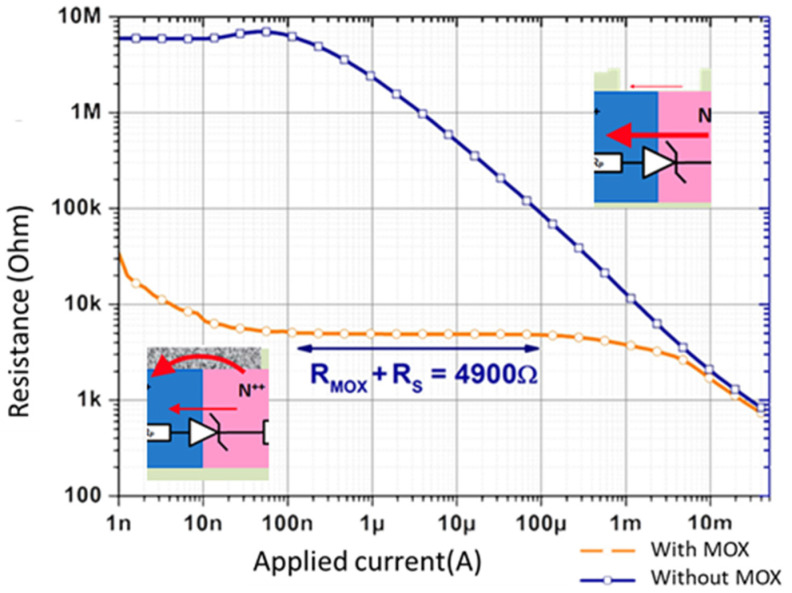
Characterization of the sensing diode with and without MOX layer.

**Figure 11 micromachines-12-01355-f011:**
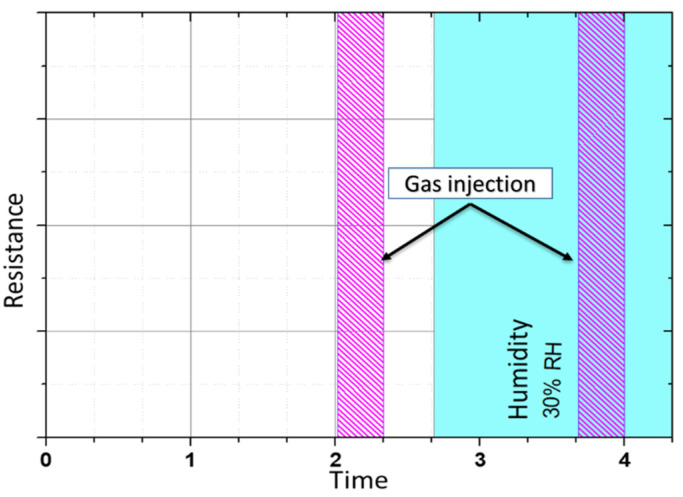
Static test protocol: equivalent resistance measurement versus time under various atmospheres.

**Figure 12 micromachines-12-01355-f012:**
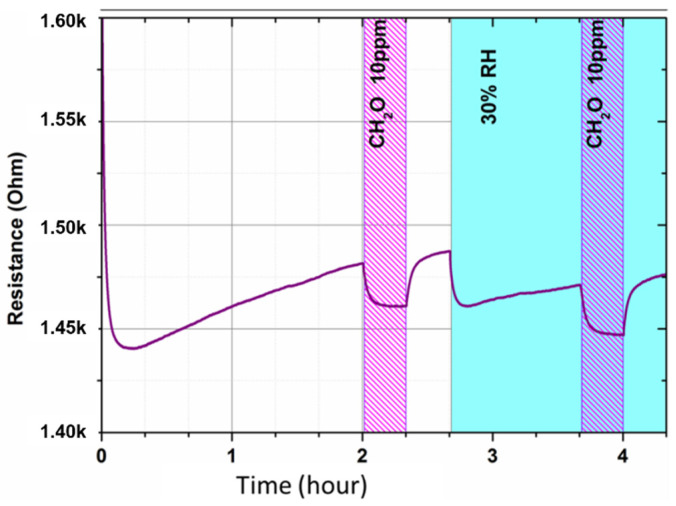
Static test: resistance measurement versus time under various atmospheres. Injection of 10 ppm of formaldehyde under dry air and then under humid air (30%RH).

**Figure 13 micromachines-12-01355-f013:**
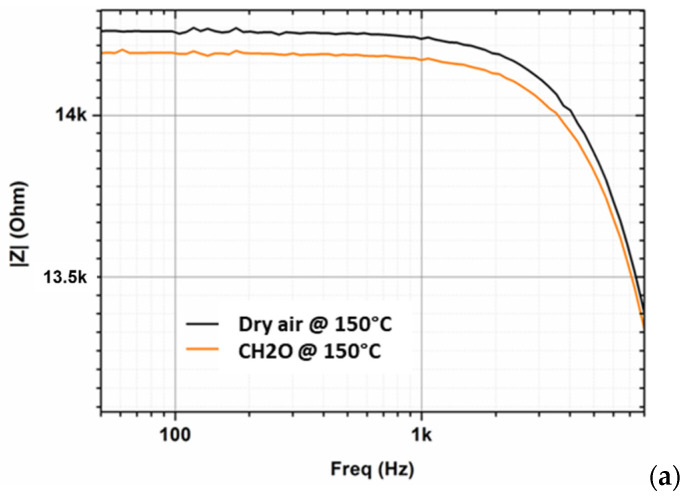
Results of dynamic tests under the same atmospheres versus in static test: (**a**) impedance at low frequencies; (**b**) resistance and capacitance at 10 kHz.

**Table 1 micromachines-12-01355-t001:** Deposition parameters of thin layers.

Target Material	ZnO:Ga
Magnetron	Yes
Power	30 W
Argon pressure	2 Pa
Target to substrate distance	7 cm
Thickness	25–50 nm

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
