# Peer review of "A New Miniaturized Gas Sensor Based on Zener Diode Network Covered by Metal Oxide"

_micromachines, 2021, doi:10.3390/mi12111355_

Round 1

Reviewer 1 Report

The article "A new miniaturized gas sensor based on Zener diode network covered by metal oxide" is devoted to obtaining gas sensors with a new design. The work can be published in Micromachines after making minor changes:

1) The introduction to this article is quite complete and interesting, nevertheless, the authors refer only to 9 references, which is insufficient for a full-fledged article. It is necessary to supplement this section with other links.

2) A lot of abbreviations are used in the work, perhaps the authors need to decipher them at the beginning of the article.

3) The authors provide a micrograph of an AFM film of ZnO, it is necessary to provide data on the dispersion and other characteristics that can be calculated using this method (roughness and others). What method was used to determine the roughness of the film. What is the error?

4) It is not entirely clear from the text of the article when the authors talk about "doping level of the P + zone ...". How does doping take place? Due to the different Ga content in ZnO? It is necessary to describe this moment in more detail, if there were several samples, then you need to write about this in the experimental part.

5) Perhaps the following articles will be useful for citation in this work: doi.org/10.1016/j.mseb.2021.115233 and doi.org/10.1016/j.ceramint.2019.11.279.

I would like to note that from time to time in the article, in different sections, various technological parameters or measurement conditions are given. These data should be presented in the experiment section. Also, pictures can be grouped so that all graphic data is in one place.

Author Response

Thank you so much for your comments and recommendations.

Point 1: References in introduction

Response 1: Thanks to your advises, we added few references of other articles that deal with low power consumption and miniaturized devices

Point 2: abbreviations

Response 2: All the abbreviations have been explained directly in the text (line 33-35).

Point 3: Roughness

Response 3: The grain size and the roughness were added. The description of the AFM experimental setup has been also added (line 194 to 206)

Point 4: Doping level of P+ zone

Response 4: The P+ zone concerns the polysilicon and not the ZnO:Ga sensitive layer. The device integrates two P+/N+ junctions (Zener diodes) head to tail made in a layer of polysilicon. The electrical behavior of these diodes (Zener voltage) is essentially driven by the lowest doping level of the p-type region (in our case) that’s why we decided to characterize different doping level at this stage. This p-type doping level is obtained by Boron inplantation after the polysilicon deposition by LPCVD process. This is what we try to explain in the section 2.2 (line 147).

Point 5: references

Response 5: Thank you very much: These two references have been added in our introduction

Concerning your remark concerning parameters and or test conditions, effectively we could concentrated all in one paragraph. That’s another way of presenting results, but we preferred to separate first electrical measurements of diodes and qualitative surface effects and then, in another section, sensor measurements with other parameters taken into account.

Reviewer 2 Report

The current manuscript describes a miniaturized gas sensor using a diode network. This work is quite interesting even though the sensing performance is not so impressive. I think it fits the scope of the Micromachines. I recommend to accept the manuscript after minor revision. My comments are as follow:

  1. Literature about low power and miniaturized gas sensors should be added to the reference and the difference between these work and this work should be discussed in the introduction part . eg Sensors and Actuators B: Chemical, 2019, 301: 127067.; ACS Sensors, 2016, 1: 339-343.
  2. The baseline of the sensor seems to draft a lot. How does the baseline behave for long term use. How is the stability of the sensor over time.

Author Response

The current manuscript describes a miniaturized gas sensor using a diode network. This work is quite interesting even though the sensing performance is not so impressive. I think it fits the scope of the Micromachines. I recommend to accept the manuscript after minor revision. My comments are as follow:

  1. Literature about low power and miniaturized gas sensors should be added to the reference and the difference between these work and this work should be discussed in the introduction part . eg Sensors and Actuators B: Chemical, 2019, 301: 127067.; ACS Sensors, 2016, 1: 339-343.
  2. The baseline of the sensor seems to draft a lot. How does the baseline behave for long term use. How is the stability of the sensor over time.

Thank you so much for your comments and recommendations.

Response 1: The references proposed have been added to our paper. Effectively, they should be cited as they deal with low power microheater and small gap Ti/Pt electrodes for gas sensors. We modified some sentences in introduction to add short comments about them (line 45 and 65)

Response 2: Yes you’re wright. Our first measurements under gas are presented in this paper just to demonstrate the proof of concept. Effectively, we did not wait long enough to wait for the sensor to stabilize before the first injection of gas but we can see in figure 12 that this stabilization can be achieved after 4-5 hours as for standard MOX sensors when they are started. Of course, some tests are in progress to confirm the way to enhance sensitivity and stability of our device knowing that these first results have been obtained at room temperature.